# BANDIT LEARNING IN MATCHING MARKETS WITH IN-DIFFERENCE

**Fang Kong**[*]    **Jingqi Tang**[†]    **Mingzhu Li**[‡]    **Pinyan Lu**[§]    **John C.S. Lui**[¶]    **Shuai Li**[∥]

## ABSTRACT

A rich line of recent works studies how participants in matching markets learn their unknown preferences through iterative interactions with each other. The two sides of participants in the market can be respectively formulated as players and arms in the bandit problem. To ensure market stability, the objective is to minimize the stable regret of each player. Though existing works provide significant theoretical upper bounds for players' stable regret, the results heavily rely on the assumption that each participant has a strict preference ranking. However, in real applications, multiple candidates (e.g., workers in the labor market and students in school admission) usually demonstrate comparable performance levels, making it challenging for participants (e.g., employers and schools) to differentiate and rank their preferences. To deal with the potential indifferent preferences, we propose an adaptive exploration algorithm based on arm-guided Gale-Shapley (AE-AGS). We show that its stable regret is of order $O(NK \log T/\Delta^2)$, where $N$ is the number of players, $K$ the number of arms, $T$ the total time horizon, and $\Delta$ the minimum non-zero preference gap. Extensive experiments demonstrate the algorithm's effectiveness in handling such complex situations and its consistent superiority over baselines.

## 1 INTRODUCTION

The two-sided matching market is a fundamental concept in economics and operations research (Gale & Shapley, 1962; Roth, 1984; Roth & Sotomayor, 1992; Roth & Peranson, 1999; Fleiner, 2003). It provides a formal framework to model interactions between two distinct sides of agents and has a wide range of applications such as labor markets (Kelso Jr & Crawford, 1982; Roth, 1984), school admission (Roth, 2008), house allocation (Sönmez & Ünver, 2011), and so forth. Each agent (e.g., employer) has his own preferences over the other side (e.g., workers in labor markets) and seeks to form beneficial pairings. To keep the stability of the market and thus avoid dissatisfaction of agents and future inefficiencies, a rich line of works study how to find a stable matching in the market (Gale & Shapley, 1962; Roth, 1984; Roth & Sotomayor, 1992; Kelso Jr & Crawford, 1982), among which the Gale-Shapley algorithm (Gale & Shapley, 1962) is one of the most classic one. All these works assume that agents' preferences are known as a prior.

However, prior knowledge of preferences may not always be fully certain in real-world applications. For example, employers typically cannot precisely assess a worker's abilities before they are hired. A stable matching derived from temporal preference estimation may not ensure long-term stability. With the rise of online marketplaces such as the online labor platform Upwork and the crowdsourcing platform Amazon Mechanical Turk, employers are increasingly able to learn about uncertain preferences through iterative matching processes driven by their released multiple tasks. The multi-armed bandit (MAB) is a classic model that characterizes the learning process for agents towards

---

[*]`kongf@sustech.edu.cn`. Southern University of Science and Technology

[†]`sjtu18296796824@sjtu.edu.cn`. Shanghai Jiao Tong University

[‡]`justbeme@sjtu.edu.cn`. Shanghai Jiao Tong University

[§]`lu.pinyan@mail.shufe.edu.cn`. Shanghai University of Finance and Economics; Key Laboratory of Interdisciplinary Research of Computation and Economics (SUFE), Ministry of Education

[¶]`cslui@cse.cuhk.edu.hk`. The Chinese University of Hong Kong

[∥]`shuaili8@sjtu.edu.cn`. Shanghai Jiao Tong University. Corresponding author

uncertain information (Auer et al., 2002; Lattimore & Szepesvári, 2020), also offering solutions for agents in matching markets to learn their unknown preferences.

The classic MAB model contains one player and $K$ arms. Each arm $a_j$ is associated with an unknown reward $\mu_j$. The player would learn this knowledge through iterative selections. The objective of the player is to maximize the cumulative rewards, equivalent to minimizing the cumulative regret defined as the cumulative distance between the optimal reward and received rewards. To achieve this long-horizon objective, the player faces the dilemma of exploration and exploitation. The former hopes to select the arm with fewer observed times to know the arm better, and the latter hopes to select the better-performed arms to accumulate as many rewards as possible. The explore-then-commit (ETC) (Garivier et al., 2016), upper confidence bound (UCB) (Auer et al., 2002) and Thompson sampling (TS) (Thompson, 1933) are common strategies to deal with the problem.

The bandit learning problem in matching markets recently attracted great interest in the literature, where two sides of participants can be modeled as players and arms. Players can learn their unknown preferences through interactions with arms. Das & Kamenica (2005) first introduces the framework and proposes empirical solutions. Liu et al. (2020) further gives a formal theoretical formulation and derives algorithms with theoretical guarantees on the stable regret, which is defined as the cumulative distance between the reward in a stable matching and the reward received during the interactions. In the matching market scenario, due to the interference among multiple agents, the selections of an individual player can be easily blocked, making the trade-off between exploration and exploitation more challenging. To avoid conflicts among players, Liu et al. (2020) consider the centralized setting where a central platform collects information from participants and assigns partners for players. A rich line of the following works try to improve their stable regret bound and generalize the model by considering the decentralized setting (Liu et al., 2021; Sankararaman et al., 2021; Basu et al., 2021; Maheshwari et al., 2022; Kong et al., 2022; Zhang et al., 2022; Kong & Li, 2023).

Despite the significance of the results, all existing works assume each market participant has a strict preference ranking, i.e., the preference values towards different candidates are different. However, this assumption may not be realistic. In many applications such as labor market and school admission, multiple candidates usually demonstrate similar performances, leading to ties of preference rankings. Especially in large markets, maintaining a strict preference ranking over all candidates can be extremely time-consuming and effort-intensive, while the marginal benefit of distinguishing between closely ranked candidates may be minimal. To improve the practicality and robustness of algorithms, it is crucial to deal with participants' indifferent preferences (Erdil & Ergin, 2008; Abdulkadiroğlu et al., 2009; Chen, 2012; Erdil & Ergin, 2017; Erdil & Kumano, 2019).

The state-of-the-art approaches in matching markets employ an explore-then-Gale-Shapley strategy to address the exploration-exploitation trade-off (Zhang et al., 2022; Kong & Li, 2023; Kong et al., 2024). In these methods, exploration continues until players have identified all preference gaps, after which the algorithm transits to exploitation, applying the Gale-Shapley algorithm (Gale & Shapley, 1962) to achieve stable matching. However, once two arms exhibit identical preferences, the exploration process would never stop, leading the algorithm to incur an $O(T)$ regret, where $T$ represents the total time horizon. With indifferent preferences, the algorithm faces new difficulties in balancing exploration and exploitation. Prolonged exploration could incur additional regret, while prematurely halting exploration may result in incorrect ranking estimates, leading to a non-stable matching.

In this work, we try to overcome the above challenge for the bandit learning problem in matching markets with indifference. Though existing results all assume market participants have strict preference rankings, we examine whether they can be extended to the indifference setting. As summarized in Table 1, only Liu et al. (2020) and Basu et al. (2021) can apply to indifference. However, their approaches either require knowledge of $\Delta$ or suffer exponential regret. We propose a more suitable policy to balance exploration and exploitation - an arm-guided adaptive exploration algorithm where players only explore arms that propose to them and adaptively eliminate sub-optimal arms, for both the centralized and decentralized setting. This design allows players to explore freely without the need to explicitly distinguish between exploration and exploitation processes. We show that such an algorithm achieves the stable regret of order $O(NK \log T/\Delta^2)$ where $N$ is the number of players,

$K$ is the number of arms, $T$ is the total horizon and $\Delta$ is the minimum non-zero gap[1]. Extensive experiments are conducted to show our algorithm's effectiveness and consistent advantage compared with available baselines.

Table 1: Comparisons of related results. $N$ is the number of players, $K$ is the number of arms, $\Delta$ is the minimum preference gap among all players for different arms in existing works, and is the minimum non-zero preference gap among all players for different arms if the result holds under indifference. $\rho, \epsilon$ are hyper-parameters. C and D represent centralized and decentralized settings, respectively. We use tiny font to annotate the parts of the original proof where it fails to hold under indifference and provide more details in Appendix A.

| References | Stable regret bound | Setting | Holds under indifference? |
|---|---|---|---|
| Liu et al. (2020) | $O\left(\frac{NK\log T}{\Delta^2}\right)$ | C | ✗ (Corollary 9) |
| Liu et al. (2021) | $O\left(\frac{N^5 K^2 \log^2 T}{\rho^{N^4}\Delta^2}\right)$ | D | ✗ (Lemma 8) |
| Kong et al. (2022) | $O\left(\frac{N^5 K^2 \log^2 T}{\rho^{N^4}\Delta^2}\right)$ | D | ✗ (Lemma 1) |
| Zhang et al. (2022) | $O\left(\frac{K\log T}{\Delta^2}\right)$ | D | ✗ (2nd paragraph in page 16) |
| Kong & Li (2023) | $O\left(\frac{K\log T}{\Delta^2}\right)$ | D | ✗ (Lemma 4) |
| Kong et al. (2024) | $O\left(\frac{N^2 \log T}{\Delta^2}\right)$ | C & D | ✗ (Lemma A.5) |
| Liu et al. (2020) | $O\left(\frac{K\log T}{\Delta^2}\right)$ | C (Known $\Delta$) | ✓ |
| Basu et al. (2021) | $O\left(K\log^{1+\epsilon} T + 2^{\Delta^{-2/\epsilon}}\right)$ | D | ✓ |
| **Ours** | $O\left(\frac{NK\log T}{\Delta^2}\right)$ | C & D | ✓ |

## 2 RELATED WORK

The model of two-sided matching markets has been studied for many years (Gale & Shapley, 1962; Roth, 1984; Roth & Sotomayor, 1992). The seminal work (Gale & Shapley, 1962) proposes the Gale-Shapley algorithm to compute a stable matching in the one-to-one markets. Some research has extended the algorithm to address more complex markets with different preference structures (Kelso Jr & Crawford, 1982; Roth & Sotomayor, 1992). Most of these works analyze the algorithm based on the assumption that all participants have a strict preference ranking. When participants have indifferent preferences, Irving (1994) define different levels of stability and propose algorithms to achieve them. Erdil & Ergin (2008) propose a method to improve satisfaction from a given stable matching. Abdulkadiroğlu et al. (2009) consider the strategy-proofness of the mechanism that whether participants have an incentive to deviate from the algorithm.

When market participants have uncertain preferences, Das & Kamenica (2005) first introduce the bandit model into matching markets. They propose an $\varepsilon$-greedy type algorithm and demonstrate its empirical performances. Liu et al. (2020) theoretically formulate this problem. They mainly study the centralized setting with a central platform computing the matching in each time slot. Both an ETC and UCB-type algorithm are proposed for this setting. The former achieves $O(K\log T/\Delta^2)$ regret with the knowledge of $\Delta$ and the latter achieves $O(NK\log T/\Delta^2)$. Liu et al. (2021) and Kong et al. (2022) generalize the problem to the decentralized setting, where players need to coordinate their selections to avoid invalid explorations due to conflicts. However, due to the interference of multiple agents in the decentralized markets, their algorithm suffers an exponential order of regret. To improve the learning efficiency, Sankararaman et al. (2021); Basu et al. (2021); Maheshwari et al. (2022); Wang & Li (2024) consider the setting where participants' preferences satisfy special assumptions thus the interference becomes easier. For these special markets, they provide an $O(NK\log T/\Delta^2)$ or $O(N\log T/\Delta^2)$ regret guarantee. Until recently, Zhang et al. (2022) and Kong & Li (2023) independently propose an explore-then-Gale-Shapley procedure and show an

---

[1]If all preference gaps are zero, we show our stable regret is 0 in the centralized setting and is $O(\log T)$ in the decentralized setting.

$O(K \log T / \Delta^2)$ stable regret upper bound for general markets. In all of the above works, both players and arms are assumed to have strict preference rankings and $\Delta$ is defined as the minimum preference gap among all players over different arms, which may be $0$ under indifference. Our work follows this line and considers the more general indifference setting.

A contemporary work (Lin et al., 2024) also investigates the indifference setting. They first consider the offline setting where players have known preferences and propose an $\alpha$-approximate oracle that returns a matching in which each player's utility is at least an $\alpha$ fraction of their optimal utility across all stable matchings. They further extend this oracle to the bandit setting, obtaining results on $\alpha$-approximate player-optimal stable regret. Their analysis also yields a problem-independent regret of order $O(T^{2/3})$. Different from this work, we focus on stable regret, which is defined on the least reward among all stable matchings, and do not consider approximate guarantees.

There are also other works studying the uncertain preferences in matching markets. The variants include the market where both sides of agents have unknown preferences (Pagare & Ghosh, 2023), the contextual markets where the player's preferences can be represented by the inner product between the preference vector and the arm feature (Li et al., 2022), the many-to-one markets where one side of agents can match more than one partners (Wang et al., 2022; Kong & Li, 2024; Li et al., 2024; Zhang & Fang, 2024), as well as the non-stationary markets where the preference of agents vary over time (Ghosh et al., 2022; Muthirayan et al., 2023).

## 3 PROBLEM SETTING

This section introduces the problem setting of bandit learning in matching markets with indifference. Denote $\mathcal{N} = \{p_1, p_2, \ldots, p_N\}$ as the player set and $\mathcal{K} = \{a_1, a_2, \ldots, a_K\}$ as the arm set, where $N$ and $K$ represent the number of players and arms, respectively. To ensure each player has a chance to be matched, we assume $N \leq K$ as existing works (Liu et al., 2020; 2021; Sankararaman et al., 2021; Basu et al., 2021; Zhang et al., 2022; Kong & Li, 2023; Wang & Li, 2024).

Each market participant has a preference ranking over the other side. Specifically, the preference value of player $p_i$ over arm $a_j$ can be portrayed by a real value $\mu_{i,j} \in (0, 1]$. A higher value represents more preferences, i.e., $\mu_{i,j} > \mu_{i,j'}$ implies $p_i$ prefers $a_j$ to $a_{j'}$. These preference values are unknown and need to be learned through interactive interactions with arms. It is worth noting that all existing works (Liu et al., 2020; 2021; Kong et al., 2022; Sankararaman et al., 2021; Basu et al., 2021; Zhang et al., 2022; Kong & Li, 2023; Wang & Li, 2024; Kong et al., 2024) assume the preference values over different arms are different, i.e., $\mu_{i,j} \neq \mu_{i,j'}$ for any player $p_i$ and arms $a_j, a_{j'}$. However, this assumption is often unrealistic in practical applications, as multiple arms (e.g., workers in labor markets or students in school admission scenarios) usually exhibit similar performances, making it difficult for players to explicitly differentiate their preferences. We relax this assumption by allowing indifferent preferences, i.e., the player can have the same preference values over different arms. On the other side, each arm $a_j$ also has preferences over players. Denote $\pi_{j,i}$ as the position of $p_i$ in $a_j$'s preference rankings. Arms can also have indifferent preferences over players. We use $\pi_{j,i} \prec \pi_{j,i'}$ to denote that $p_i$ has a higher ranking so is more preferred than $p_{i'}$ by $a_j$. And $\pi_{j,i} = \pi_{j,i'}$ represents $a_j$ can not distinguish the performances between $p_i$ and $p_{i'}$. Similar to the labor market scenario where workers (arms) usually have an evaluation system based on the known characteristics of the employers (players) such as the salary, location, and so forth, we assume each arm knows their own preference ranking as existing works (Liu et al., 2020; 2021; Kong et al., 2022; Sankararaman et al., 2021; Basu et al., 2021; Zhang et al., 2022; Kong & Li, 2023; Wang & Li, 2024; Kong et al., 2024).

The players would iteratively interact with the arms. At each time slot $t = 1, 2, 3, \ldots$, each player $p_i$ selects an arm $A_i(t) \in \mathcal{K} \cup \{-1\}$, where we use $-1$ to represent that $p_i$ does not select any arm in this time slot. For the arm side, each arm $a_j$ receives the proposals from $A_j^{-1}(t) = \{p_i : A_i(t) = a_j\}$. Due to the capacity constraint, it only accepts the most preferred one, i.e., the player $A_j^{-1}(t) \in \arg\min_{i \in A_j^{-1}(t)} \pi_{j,i}$ with the highest preference ranking. When there are multiple choices, the arm would randomly break the tie. For the player side, any player $p_i$ whose proposal is accepted would successfully match with $A_i(t)$. It would receive a reward $X_{i,A_i(t)}(t)$ characterizing its satisfaction over this matching experience, where we assume the reward is a $1$-subgaussian random variable with expectation $\mu_{i,A_i(t)}$ as existing bandit works (Lattimore & Szepesvári, 2020). And if $p_i$'s proposal is

rejected, it only receives $X_{i,A_i(t)}(t) = 0$. For convenience, we use $\bar{A}(t) = (\bar{A}_i(t))_{i \in [N]}$ to represent the final matching outcome in time slot $t$, where $\bar{A}_i(t) = A_i(t)$ if $p_i$ is successfully matched and $\bar{A}_i(t) = -1$ otherwise.

To ensure long-term equilibrium in the market, the players aim to find a stable matching. Given a matching $\bar{A} := (\bar{A}_i)_{i \in [N]}$, if there exists a pair $(p_i, a_j)$ such that $p_i$ prefers $a_j$ to its current partner $\bar{A}_i$ and $a_j$ also prefers $p_i$ to its current partner $\bar{A}_j^{-1}$, i.e., $\mu_{i,j} > \mu_{i,\bar{A}_i}$ and $\pi_{j,i} \prec \pi_{j,\bar{A}_j^{-1}}$, then $p_i$ and $a_j$ has the incentive to deviate from their partners. In this case, the matching $\bar{A}$ is unstable, and such a pair is called a blocking pair. A stable matching is a matching without any blocking pair. It is worth noting that there may be more than one stable matching in the market. Denote $M := \{m := (m_i)_{i \in [N]} : m \text{ is stable}\}$ as the set of all stable matchings. Existing works study the player-optimal stable matching (Liu et al., 2020; Zhang et al., 2022; Kong & Li, 2023; Kong et al., 2024) which is defined as the stable matching in which all players are matched with their most preferred arm among all stable matchings and the player-pessimal stable matching (Liu et al., 2020; 2021; Kong et al., 2022) which is defined as the stable matching in which all players are matched with their least preferred arm among all stable matchings. However, when the market participants have indifferent preferences, such two stable matchings may not exist. Example 3.1 illustrates one possible case.

**Example 3.1.** *The market contains 3 players and 3 arms with the preference rankings listed below:*

$$\begin{cases} p_1 : a_1 = a_2 \succ a_3 \,, \\ p_2 : a_1 \succ a_2 = a_3 \,, \\ p_3 : a_1 \succ a_2 \succ a_3 \,, \end{cases} \quad \begin{cases} a_1 : p_1 \succ p_2 = p_3 \,, \\ a_2 : p_1 \succ p_2 \succ p_3 \,, \\ a_3 : p_1 \succ p_2 \succ p_3 \,, \end{cases}$$

*where $a_2 \succ a_3$ for $p_1$ implies $p_1$ prefers $a_2$ over $a_3$. In this market, both $\{(p_1, a_2), (p_2, a_1), (p_3, a_3)\}$ and $\{(p_1, a_2), (p_2, a_3), (p_3, a_1)\}$ are stable matchings. But players $p_2$ and $p_3$ do not match with the most preferred arm in a common stable matching.*

In this work, we focus on the stable regret of each player $p_i$ which is defined as the difference between the least reward $\mu_{i,m_i} = \min_{m' \in M} \mu_{i,m_i'}$ that can be obtained in any stable matching and the reward accumulated during the interaction process, i.e.,

$$Reg_i(T) = \mathbb{E}\left[\sum_{t=1}^{T} (\mu_{i,m_i} - X_{i,A_i}(t))\right] , \tag{1}$$

where the expectation is taken from the randomness of the reward and players' policies[2].

## 4 ALGORITHM IN THE CENTRALIZED SETTING

In this section, we introduce our proposed adaptive exploration with arm-guided GS (AE-AGS) algorithm. To better convey the algorithm idea, we first present the centralized version (Algorithm 1) where a central platform collects information from market participants and computes the matching.

---

**Algorithm 1** adaptive exploration with arm-guided GS (AE-AGS, centralized version, from the view of the central platform)

---

1: **for** time slot $t = 1, 2, \ldots$ **do**
2:     Collect the arms' preference rankings $(\pi_{j,i})_{i \in [N]}$ from each arm $a_j \in \mathcal{K}$
3:     Collect the matched times $(T_{i,j})_{j \in [K]}$ and the comparison matrix $(\text{Better}(i, j, j'))_{j,j' \in [K]}$ from each player $p_i \in \mathcal{N}$
4:     Compute $A(t) = \text{Subroutine-of-AE-AGS}(\pi_{j,i}, T_{i,j}, \text{Better}(i, j, j'))_{i \in [N], j, j' \in [K]}$
5:     Assign the arm $A_i(t)$ to each player $p_i \in \mathcal{N}$
6: **end for**

---

[2]It is worth noting that the worst partner for all players may not appear in a single stable matching. As demonstrated in the proof, we bound the stable regret by constraining the number of unstable matchings. So our stable regret upper bound also applies to the cumulative market instability (the cumulative number of unstable matchings).

Specifically, in each time slot $t$, each arm $a_j$ would submit its preference ranking $(\pi_{j,i})_{i\in[N]}$ to the central platform (Line 2). If multiple players share the same preferences, the arm can randomly break the tie. And each player $p_i$ maintains a counter $T_{i,j}$ representing the number of times that $p_i$ is matched with arm $a_j$. It also maintains a comparison matrix Better among each arm pair. $\text{Better}(i,j,j') = 1$ means $p_i$ estimate that it prefers $a_j$ over $a_{j'}$. And $\text{Better}(i,j,j') = 0$ means $p_i$ still cannot distinguish the performances between $a_j$ and $a_{j'}$, or estimates that it prefers $a_{j'}$ over $a_j$. The player would submit the counter and comparison matrix to the central platform (Line 3).

Then, the central platform would compute a matching $A(t)$ based on the collected information (Line 4) and assign the target arm $A_i(t)$ to player $p_i$ (Line 5). The detailed procedure to compute $A(t)$ is summarized in the Subroutine-of-AE-AGS algorithm (Algorithm 2). In general, Algorithm 2 can be regarded as an adaptive exploration algorithm based on GS with the arm side as the proposing side. Arms would propose to their most preferred players based on their submitted preference ranking (Line 3). Among the received proposals, players would first compute the estimated sub-optimal arms, i.e., an arm $a_j$ can be regarded as sub-optimal if there exists a player $a_{j'}$ such that $p_i$ determines it prefers $a_{j'}$ over $a_j$ (Line 5). Each player $p_i$ would accept the proposal from the potential optimal arm with the least matched times (Line 6). If the arm is not accepted by its proposed player, it proposes to the next preferred player (Line 8). Until all arms are matched or have proposed all of the $N$ players (Line 2), the algorithm stops and outputs the final matching. It is worth noting that Algorithm 2 ensures that all players are assigned different arms when stopping since $N \leq K$.

---

**Algorithm 2** Subroutine-of-AE-AGS

**Input:** Arms' preference rankings $(\pi_{j,i})_{j\in[K],i\in[N]}$, player-arm matched times $(T_{i,j})_{i\in[N],j\in[K]}$, comparison matrix $(\text{Better}(i,j,j'))_{i\in[N],j,j'\in[K]}$
1: **Initialize:** $\forall p_i \in \mathcal{N}$, its available arm set $\mathcal{A}_i = \emptyset$, temporarily matched arm $m_i = -1$;
   $\forall a_j \in \mathcal{K}$, its current proposing ranking $s_j = 1$, temporarily matched player $m_j^{-1} = -1$
2: **while** $\exists a_j : m_j^{-1} = -1$ and $s_j \leq N$ **do**
3:   Denote $p_i$ as the player who ranked at the position $s_j$, i.e., $p_i := \pi_{j,s_j}$
4:   Update the available arm set: $\mathcal{A}_i = \mathcal{A}_i \cup \{a_j\}$
5:   Compute the estimated sub-optimal arm set in $\mathcal{A}_i$:
     $\mathcal{D}_i = \{a_j \in \mathcal{A}_i : \exists j' \in \mathcal{A}_i \text{ s.t. } \text{Better}(i,j',j) = 1\}$
6:   Update the temporarily matched arm of $p_i$ as $m_i \in \arg\min_{j\in\mathcal{A}_i\setminus\mathcal{D}_i} T_{i,j}$
     Suppose $a_k$ is the temporarily matched arm of $p_i$, i.e., $a_k = m_i$, update $m_k^{-1} = p_i$
7:   **for** $a_{j'} \in \mathcal{A}_i$ and $a_{j'} \neq m_i$ **do**
8:     $s_{j'} = s_{j'} + 1, m_{j'}^{-1} = -1$
9:   **end for**
10: **end while**
**Output:** Matching outcome $m = (m_i)_{i\in[N]}$

---

The operation of players is summarized in Algorithm 3. Each player $p_i$ maintains $\hat{\mu}_{i,j}$ and $T_{i,j}$ to record the estimated preference value and the matched time with arm $a_j$ (Line 1). In each time slot $t$, the player $p_i$ first computes the upper confidence bound $\text{UCB}_{i,j}$ and lower confidence bound $\text{LCB}_{i,j}$ as Line 3. It can be shown in the analysis that the real preference value $\mu_{i,j}$ can be upper bounded by $\text{UCB}_{i,j}$ and lower bounded by $\text{LCB}_{i,j}$ with high probability. So once an arm $a_j$'s lower bound is better than the other arm $a_{j'}$'s upper bound, $p_i$ can regard it prefers $a_j$ over $a_{j'}$ and update $\text{Better}(i,j,j') = 1$ (Line 4). Each player would then submit the information of matched times and comparison matrix to the central platform (Line 5) and receive the assigned target arm $A_i(t)$ (Line 6). It then selects this arm and updates the estimated preference values and matched times based on the received rewards (Line 7-9).

## 4.1 THEORETICAL RESULTS

This section provides the theoretical results for the centralized AE-AGS algorithm. To characterize the hardness of the learning process, we first define the preference gap $\Delta$ as follows.

**Definition 4.1.** For any player $p_i$ and arm $a_j, a_{j'}$, define $\Delta_{i,j,j'} = |\mu_{i,j} - \mu_{i,j'}|$ as the preference gap of $p_i$ between $a_j$ and $a_{j'}$. Further, define $\Delta = \min_{i,j,j',\Delta_{i,j,j'}\neq 0} \Delta_{i,j,j'}$ as the minimum non-zero gap if $\min_{i,j,j',\Delta_{i,j,j'}\neq 0} \Delta_{i,j,j'} \neq 0$. Otherwise, define $\Delta = 0$.

---

**Algorithm 3** AE-AGS (centralized version, from the view of player $p_i$)

---

1: **Initialize:** $\forall j \in [K], \hat{\mu}_{i,j} = 0, T_{i,j} = 0$
$\forall j, j' \in [K], \text{Better}(i, j, j') = 0$ // $\text{Better}(i, j, j') = 1$ implies that $p_i$ considers that $a_j$ is better than $a_{j'}$, $\text{Better}(i, j, j') = 0$ otherwise
2: **for** time slot $t = 1, 2, \ldots$ **do**
3:     Compute the upper and lower confidence bounds for each arm $a_j \in \mathcal{K}$ as
    $\text{UCB}_{i,j} = \hat{\mu}_{i,j} + \sqrt{6 \log T / T_{i,j}}, \text{LCB}_{i,j} = \hat{\mu}_{i,j} - \sqrt{6 \log T / T_{i,j}}$
    // If $T_{i,j} = 0$, then $\text{UCB}_{i,j} = \infty, \text{LCB}_{i,j} = -\infty$
4:     Update Better for any $j, j' \in [K]$: $\text{Better}(i, j, j') = 1$ if $\text{LCB}_{i,j}(t) > \text{UCB}_{i,j'}(t)$
5:     Submit $(T_{i,j})_{j \in [K]}, (\text{Better}(i, j, j'))_{j,j' \in [K]}$ to the central platform
6:     Receive $A_i(t)$ from the central platform and select this arm, receive reward $X_{i,A_i(t)}(t)$
7:     **if** $p_i$ is successfully accepted by $A_i(t)$ **then**
8:         $\hat{\mu}_{i,A_i(t)} = (X_{i,A_i(t)}(t) + \hat{\mu}_{i,A_i(t)} \cdot T_{i,A_i(t)})/(T_{i,A_i(t)} + 1), T_{i,A_i(t)} = T_{i,A_i(t)} + 1$
9:     **end if**
10: **end for**

---

The stable regret by following the centralized AE-AGS algorithm can be bounded as follows.

**Theorem 4.1.** *Following Algorithm 1 and 3, if $\Delta > 0$, the stable regret of each player $p_i$ satisfies*

$$Reg_i(T) \leq O(NK \log T / \Delta^2).$$

*If $\Delta = 0$, the stable regret of each player $p_i$ is $Reg_i(T) = 0$.*

Due to the space limit, the detailed proof is deferred to Appendix B. The algorithm can also be extended to the decentralized setting. We provide more discussions regarding its implementation, problem challenge, and the corresponding theoretical results in the next section.

The experiments are deferred to Appendix E.

## 5 DECENTRALIZED SETTING

In real applications, the central platform may not be always available. For generality, we also extend the AE-AGS algorithm to the decentralized setting. In this case, we follow existing decentralized works (Liu et al., 2021; Kong et al., 2022; Kong & Li, 2023) and assume that each player can observe the successfully matched pairs in each time slot. This is also common in real applications such as the workers usually updating their online profile in the market and the schools usually publishing the admission list. The decentralized version of the algorithm is presented in Algorithm 4. Due to that the algorithm proceeds in several phases, we use $\tau$ as the local time slot index during each phase.

To avoid conflicts among players when selecting arms, Algorithm 4 starts from an index estimation phase where each player learns a unique index that guides the following selections. The players can simultaneously learn arms' preference rankings during this phase (Line 3-10). Specifically, the phase contains $NK$ rounds and each arm corresponds to a $N$-round block. At the first round in arm $a_j$'s $N$-round block, all players would first select arm $a_j$. The successfully accepted player can be regarded as ranked in the first position in $a_j$'s ranking and receives an index of 1. In each of the following time $\tau$, the previously accepted players would not select arms and only the previously rejected players select arm $a_j$. The accepted one is regarded as ranked in the $\tau$-th position and receives an index $\tau$. Then after $NK$ rounds, each player knows all arms' preference rankings and gets a unique index.

The algorithm then enters the main exploration part. The total horizon can be further divided into several phases (Line 12) with the phase length growing exponentially if no player breaks the process (Line 13). Within the phase, each player locally runs the Subroutine-of-AE-AGS (Algorithm 2) with its local knowledge of all arms' preferences, player-arm matched times, and the comparison matrix (Line 15). The player then selects the computed target arm (Line 16), receives the reward, and updates its estimated preference value (Line 17). The player also updates its local counter $T_{i,j}$ for the observed matched player-arm pair $(p_i, a_j)$.

When the phase ends, players will update their comparison information based on the previous reward observations (Line 20-24). Specifically, $p_i$ uses $\text{Update\_Flag}_i(s)$ to indicate whether it has updated

---

**Algorithm 4** AE-AGS (decentralized version, from the view of player $p_i$)

---

1: **Initialize:** $\text{Better}(i, j, j') = 0, T_{i,j} = 0, \forall i \in [N], j, j' \in [K]; \hat{\mu}_{i,j} = 0, \forall j \in [K]$
   $\text{Update\_Flag}(0) = \text{False}$ // $\text{Update\_Flag} = \text{False}$ means no player updates the $\text{Better}$ matrix,
   $\text{Update\_Flag} = \text{True}$ otherwise
2: **Initialize:** $\pi_{j,i} = -1, \text{Index}_i = -1, \forall j \in [K], i \in [N]$
3: **for** $j \in [K]$ **do**
4:       $\text{Arm} = a_j$
5:       **for** round $\tau = 1, 2, \cdots, N$ **do**
6:           $A_i(\tau) = \text{Arm}$
7:           Set $\text{Arm} = -1$ and $\text{Index}_i = \tau$ if accepted by $A_i(\tau)$
8:           Update $\pi_{j,\bar{A}_j^{-1}(\tau)} = \tau$
9:       **end for**
10: **end for**
11: $\ell_0 = 2$ // The length of the phase
12: **for** phase $s = 1, 2, \cdots$ **do**
13:      $\ell_s = 2\ell_{s-1}$ if $\text{Update\_Flag}(s-1) = \text{False}$ and $\ell_s = 2$ otherwise
14:      **for** round $\tau = 1, 2, \cdots, \ell_s$ **do**
15:          $m = \text{Subroutine-of-AE-AGS}(\pi_{j,i}, T_{i,j}, \text{Better}(i, j, j'))_{i \in [N], j, j' \in [K]}$
16:          Select arm $A_i(\tau) = m_i$
17:          Update the empirical mean $(\hat{\mu}_{i,j})_{j \in [K]}$ as Line 7-9 in Algorithm 3
18:          For each arm $a_j$, observe its matched player $\bar{A}_j^{-1}(\tau)$ and update $T_{\bar{A}_j^{-1}(\tau),j} += 1$
19:      **end for**
20:      $\text{Update\_Flag}_i(s) = \text{False}, \text{Update\_Pairs}_i(s) = \{\}$
21:      **for** $j, j' \in [K]$ and $\text{Better}(i, j, j') = 0$ and $\text{UCB}_{i,j'} < \text{LCB}_{i,j}$ **do**
22:          $\text{Update\_Flag}_i(s) = \text{True}$
23:          $\text{Update\_Pairs}_i(s).\text{add}((j, j'))$
24:      **end for**
25:      $\text{Update\_Flag}(s), \text{Better} = \text{Communication}(\text{Update\_Flag}_i(s), \text{Update\_Pairs}_i(s), \text{Better})$
26: **end for**

---

the comparison information in the phase $s$, and uses $\text{Update\_Pairs}_i(s)$ to restore the updated pairs, where a pair $(j, j')$ is included in $\text{Update\_Pairs}_i(s)$ if $p_i$ identifies that $\text{LCB}_{i,j} > \text{UCB}_{i,j'}$ at the end of phase $s$. Then players communicate the updated information with each other through the $\text{Communication}$ procedure (Line 25). After communicating with others, players get the $\text{Update\_Flag}(s)$ that represents whether a player has updated his comparison information and the updated $\text{Better}$ matrix. If $\text{Update\_Flag}(s)$ is true, the players may need to explore some new arms, so the phase length must be restarted to avoid additional exploration cost (Line 13).

The detailed $\text{Communication}$ description is presented in Algorithm 5. Generally speaking, players would transmit their information one by one based on their unique index. In the first round, each player would select the arm with its own index if its $\text{Update\_Flag}$ is true and select nothing otherwise (Line 2-4). So for other players, if they observe that an arm $a_j$ is matched in this round, they can infer that the player with index $j$ has updated its comparison information in this phase and would transmit the updated pairs in the following. The following rounds can then be divided into $N$ blocks where the $p$-th block is used for player with index $p$ to transmit information and others to receive the information from this player (Line 6). If the player has no information to update, then the block can be regarded as having 0 round (Line 7-8). Otherwise, the player would select the arm in its $\text{Update\_Pairs}_i$ one by one (Line 9-15) with a round selecting nothing indicating the end of the transmission (Line 16). And other players would receive the updated pairs by observing the successfully matched arms in the corresponding block (Line 19-26). After the communication, the player gets $\text{Flag}$ that represents whether a player updates its comparison information in this phase as well as the updated $\text{Better}$ matrix. The communication procedure ensures that all players locally maintain the up-to-date comparison information of all players.

---

**Algorithm 5** `Communication`

---

**Input:** $\text{Update\_Flag}_i$, $\text{Update\_Pairs}_i$, Better
 1: **Initialize:** Flag = False, $\tau = 1$
 2: **if** $\text{Update\_Flag}_i$ = True **then**
 3:     Select arm $A_i(\tau) = a_{\text{Index}_i}$
 4: **end if**
 5: $p = 1$     // the player index who transmit information currently
 6: **while** $p \leq N$ **do**
 7:     **if** $a_p$ is not matched at time slot $\tau = 1$ **then**
 8:         $p = p + 1$
 9:     **else if** $a_p$ is matched at time slot $\tau = 1$ and $p = \text{Index}_i$ **then**
10:         Flag = True
11:         **for** $(j, j') \in \text{Update\_Pairs}_i$ **do**
12:             $\tau = \tau + 1$, $A_i(\tau) = a_j$
13:             $\tau = \tau + 1$, $A_i(\tau) = a_{j'}$
14:             Update $\text{Better}(i, j, j') = 1$
15:         **end for**
16:         $\tau = \tau + 1$, $A_i(\tau) = -1$
17:         $p = p + 1$
18:     **else**
19:         Flag = True
20:         Denote $p_{i'}$ as the player with index $p$
21:         $\tau = \tau + 1$
22:         **while** $\bar{A}_{i'}(\tau) \neq -1$ **do**
23:             $j := \bar{A}_{i'}(\tau)$, $\tau = \tau + 1$
24:             $j' := \bar{A}_{i'}(\tau)$, $\tau = \tau + 1$
25:             Update $\text{Better}(i', j, j') = 1$
26:         **end while**
27:         $p = p + 1$
28:     **end if**
29: **end while**
**Output:** Flag, Better

---

## 5.1 THEORETICAL RESULTS AND DISCUSSIONS

Algorithm 4 is a decentralized version of Algorithm 1. Compared with that in the centralized version, the algorithm only pays additional regret for index estimation and communication, which only costs a constant number of time slots and does not influence the regret order.

**Theorem 5.1.** *Following Algoirthm 4, if $\Delta > 0$, the stable regret of each player $p_i$ satisfies*

$$Reg_i(T) \leq O(NK \log T/\Delta^2).$$

*If $\Delta = 0$, the stable regret of each player $p_i$ satisfies $Reg_i(T) = O(\log T)$.*

Due to the space limit, the proof of Theorem 5.1 is deferred to Appendix C. How to balance exploration and exploitation is important to achieving lower stable regret. The state-of-the-art works (Zhang et al., 2022; Kong & Li, 2023; Kong et al., 2024) in matching markets distinctly separate exploration from exploitation, where players only shift to exploitation once the preferences for arms have been clearly differentiated. Assuming all preference values are distinct, players can keep exploring until all gaps are identified. However, under indifference, when a player cannot differentiate between two arms, it becomes challenging to discern whether further exploration is necessary. Continuous exploration may bring higher regret when preferences are the same; while discontinuing exploration may result in insufficient observations to identify preference differences and further exploiting a non-stable matching. The key to learning under indifference, therefore, is to allow players to explore without the burden of suffering additional regret. Though Liu et al. (2020) and Basu et al. (2021) can be extended to handle indifference, they either use the value of $\Delta$ to control the exploration budget (Liu et al., 2020), or adopt exponential time as the trial-and-error cost to avoid prematurely exploiting a non-stable matching (Basu et al., 2021). This results in their algorithms requiring strong assumptions and suffering from exponential regret.

Our approach provides a more adaptive perspective to balance exploration and exploitation under indifference. Players only need to explore arms that propose to them. If these arms share the same preferences, all become potential partners in a stable matching, making exploration cost-free and preserving the opportunity to exploit the stable outcome. If the arms have different preferences, the player will eventually eliminate suboptimal options after collecting sufficient observations. Such a design prevents players from deciding when to stop exploring and naturally addresses the learning challenge under indifference.

Although the ODA algorithm in Kong & Li (2024) and AE arm-DA algorithm in Hosseini et al. (2024) are also inspired by the arm-guided GS, they differ fundamentally from our approach in exploration-exploitation design principles. These two algorithms still adopt an "explore-then-exploit" strategy, explicitly dividing each step of the GS process. Only when the player completes an exploration step and identifies the optimal arm does the process move to the next step. Such an approach is still unable to handle indifferent preferences, which could lead to the algorithm getting stuck in one of the steps. Our key idea to address indifferences is to prevent players from facing the dilemma of determining when to stop exploration. In our approach, the available arms for players in each round are determined dynamically as the outcome of a multi-step GS process, allowing players to freely explore. Our convergence results also differ from Kong & Li (2024); Hosseini et al. (2024). These two algorithms converge to a fixed stable matching when players have strict preferences but may fail to converge under indifference. In contrast, our algorithm can guarantee stability under indifferences, with outcomes potentially switching between different stable matchings.

## 6 CONCLUSION

In this work, we study the bandit learning problem in more general matching markets with indifference. Under this setting, the exploration-exploitation strategies employed by existing algorithms become ineffective. To enable players to explore unknown arms without incurring significant costs, we propose a novel adaptive exploration strategy based on the arm-guided GS algorithm. This approach allows players to freely explore arms with indistinguishable preferences while ensuring efficient exploitation of stable matchings. We prove that the algorithm achieves a stable regret bound of $O(NK \log T / \Delta^2)$. We also analyze existing algorithms and demonstrate their limitations when extended to handle indifference. Compared with the two existing algorithms that can be extended to indifference, our method shows a significant improvement with respect to not only the assumptions but also the regret order. The convergence and effectiveness of our algorithm are further validated through a series of experiments.

One future direction is to explore stronger objectives. There are various levels of stability under indifference, including stronger stability, super stability, and weak stability (Irving, 1994). This work focuses on weak stability, which aligns with existing research on strict preference settings. Investigating stronger objectives is an interesting avenue for further study. Under weak stability, while a player-optimal stable matching may not always exist, it may be possible to establish guarantees for Pareto-efficient stable matchings—ensuring that no player is matched with a better arm in comparison to another stable matching while all other players are matched with an arm that is no worse (Erdil & Ergin, 2008).

## ACKNOWLEDGEMENTS

Fang Kong is supported by Guangdong Basic and Applied Basic Research Foundation. The corresponding author Shuai Li is supported by National Key Research and Development Program of China (2022ZD0114804) and National Natural Science Foundation of China (62376154). The work of John C.S. Lui is supported in part by the RGC GRF-14202923. Pinyan Lu is supported by National Key R&D Program of China (2023YFA1009500).

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

## A    DISCUSSION ON THE SUCCESS OR FAILURE OF EXISTING ALGORITHMS WHEN DEALING WITH INDIFFERENCE

In this section, we try to extend existing algorithms for general one-to-one markets (Liu et al., 2020; 2021; Basu et al., 2021; Zhang et al., 2022; Kong & Li, 2023; Kong et al., 2024) to the indifference setting. We specify the failure parts of the original proof if it cannot work under indifference and a sketched reason for Liu et al. (2020) and Basu et al. (2021) to deal with indifference.

For the centralized UCB algorithm in Liu et al. (2020), Corollary 9 does not hold as when players have indifferent preferences, $\sum_{j':\mu_{i,j'}<\mu_{i,j}} \leq \sum_{\ell=1}^{K} 1/(\ell^2 \Delta^2)$ does not hold.

For the CA-UCB algorithm in Liu et al. (2021), Lemma 8 does not hold. We can provide a counterexample that $m_t$ is stable and $E_{t+1}$ holds, but $m_{t+1} \notin M^*$. For example, there are 3 players and 3 arms with preference rankings list below:

$$\begin{cases} p_1 : a_1 \succ a_2 = a_3\,, \\ p_2 : a_2 = a_1 \succ a_3\,, \\ p_3 : a_1 = a_3 \succ a_2\,, \end{cases} \qquad \begin{cases} a_1 : p_2 \succ p_3 \succ p_1\,, \\ a_2 : p_1 \succ p_2 \succ p_3\,, \\ a_3 : p_1 \succ p_2 \succ p_3\,. \end{cases}$$

At $t$, the matching $m_t = \{(p_1, a_3), (p_2, a_2), (p_3, a_1)\}$ is stable. And at time $t + 1$, the matching can be $\{(p_1, a_2), (p_2, a_1), (p_3, -1)\}$, which is unstable. The same example can illustrate the failure of Lemma 1 in Kong et al. (2022).

For the ML-ETC algorithm in Zhang et al. (2022), the second paragraph in page 16 does not hold as when players have indifferent preferences, there always exists a pair of arms such that the stopping condition is never satisfied (the last paragraph in page 6). Similarly, for the ETGS algorithm in Kong & Li (2023), Lemma 4 does not hold as it may never happen that a pair of arms with the LCB of one is better than the UCB of the other when their preference values are the same. The same analysis applies to Lemma A.5 in Kong et al. (2024).

The proof of the centralized ETC algorithm in Liu et al. (2020) and Basu et al. (2021) goes through under indifference with $\Delta$ defined as the minimum non-zero preference gap among all players. The reason is that when the matched time of players over arms is enough to identify the minimum non-zero gap $\Delta$, the matching process in these two algorithms can be regarded as running the offline GS algorithm by randomly breaking the tie, resulting in the stable matching.

## B    PROOF OF THEOREM 4.1

For convenience, for any time slot $t$, define $\hat{\mu}_{i,j}(t), T_{i,j}(t), \mathrm{LCB}_{i,j}(t), \mathrm{UCB}_{i,j}(t)$ as the value of $\hat{\mu}_{i,j}, T_{i,j}, \mathrm{LCB}_{i,j}, \mathrm{UCB}_{i,j}$ in the AE-AGS algorithm at the start of $t$. Define the failure event

$$\mathcal{F} = \left\{ \exists i \in [N], j \in [K], t \in [T] : |\hat{\mu}_{i,j}(t) - \mu_{i,j}| > \sqrt{\frac{6\log T}{T_{i,j}(t)}} \right\} \tag{2}$$

to represent that some estimated preference value is far from the real preference value at some round $t$. When $\Delta > 0$, the stable regret of Algorithm 1 and Algorithm 3 can be decomposed as

$$Reg_i(T) = \mathbb{E}\left[\sum_{t=1}^{T} \left(\mu_{i,m_i} - X_{i,A_i(t)}(t)\right)\right]$$

$$\leq \mathbb{E}\left[\sum_{t=1}^{T} \mathbb{1}\{\bar{A}(t) \notin M\}\right]$$

$$= \mathbb{E}\left[\sum_{t=1}^{T} \mathbb{1}\{A(t) \notin M\}\right] \tag{3}$$

$$\leq \mathbb{E}\left[\sum_{t=1}^{T} \mathbb{1}\{A(t) \notin M\} \,|\, \neg\mathcal{F}\right] + \mathbb{E}\left[\sum_{t=1}^{T} \mathbb{1}\{\mathcal{F}\}\right]$$

$$\leq \frac{96NK \log T}{\Delta^2} + 2NK\,, \tag{4}$$

where equation 3 holds since all players select different arms in the centralized setting and thus no rejection happens, equation 4 is because of Lemma B.1 and Lemma B.2.

When $\Delta = 0$, all players have the same preferences over all arms. So any matching that each player is matched with an arm is a stable matching since no blocking pair exists. Since Algorithm 1 assigns different arms to different players, the matching $A(t)$ in each time slot is a stable matching. So the stable regret of all players is 0 as equation 3 is 0.

**Lemma B.1.**

$$\mathbb{E}\left[\sum_{t=1}^{T} \mathbb{1}\{\mathcal{F}\}\right] \leq 2NK \,. \tag{5}$$

*Proof.* Recall that $\mathcal{F}$ is defined as equation 2. Then,

$$
\begin{aligned}
\mathbb{E}\left[\sum_{t=1}^{T} \mathbb{1}\{\mathcal{F}\}\right] &= \mathbb{E}\left[\sum_{t=1}^{T} \mathbb{1}\left\{\exists i \in [N], j \in [K], t \in [T] : |\hat{\mu}_{i,j}(t) - \mu_{i,j}| > \sqrt{\frac{6\log T}{T_{i,j}(t)}}\right\}\right] \\
&\leq T \cdot \sum_{i\in[N]}\sum_{j\in[K]} \mathbb{E}\left[\sum_{t=1}^{T} \mathbb{1}\left\{|\hat{\mu}_{i,j}(t) - \mu_{i,j}| > \sqrt{\frac{6\log T}{T_{i,j}(t)}}\right\}\right] \\
&= T \cdot \sum_{i\in[N]}\sum_{j\in[K]}\sum_{t=1}^{T}\sum_{\omega=1}^{t} \mathbb{P}\left(T_{i,j}(t) = \omega, |\hat{\mu}_{i,j}(t) - \mu_{i,j}| > \sqrt{\frac{6\log T}{T_{i,j}(t)}}\right) \\
&= T \cdot \sum_{i\in[N]}\sum_{j\in[K]}\sum_{t=1}^{T}\sum_{\omega=1}^{t} \mathbb{P}\left(|\hat{\mu}_{i,j,\omega} - \mu_{i,j}| > \sqrt{\frac{6\log T}{\omega}}\right) \\
&\leq T \cdot \sum_{i\in[N]}\sum_{j\in[K]}\sum_{t=1}^{T}\sum_{\omega=1}^{t} 2\exp(-3\log T) \tag{6} \\
&\leq 2NK \,.
\end{aligned}
$$

Here equation 6 is due to Lemma D.1. □

**Lemma B.2.** *Following Algorithm 1 and Algorithm 3, when $\Delta > 0$, it holds that*

$$\mathbb{E}\left[\sum_{t=1}^{T} \mathbb{1}\{A(t) \notin M\} \,|\, \neg\mathcal{F}\right] \leq \frac{96NK\log T}{\Delta^2} \,. \tag{7}$$

*Proof.* For convenience, denote $\mathcal{A}_i(t)$ as the set of available arms of player $i$ at the end of Algorithm 2 when running it at round $t$. According to the definition of stable matching, we can first decompose

the above regret as

$$
\mathbb{E}\left[\sum_{t=1}^{T}\mathbb{1}\{A(t)\notin M\}\mid\neg\mathcal{F}\right]
$$

$$
\leq\mathbb{E}\left[\sum_{t=1}^{T}\mathbb{1}\left\{\exists i\in[N], j\in[K]: \mu_{i,j}>\mu_{i,A_i(t)} \text{ and } \pi_{j,i}\prec\pi_{j,A_j^{-1}(t)}\right\}\mid\neg\mathcal{F}\right]
$$

$$
\leq\mathbb{E}\left[\sum_{t=1}^{T}\mathbb{1}\left\{\exists i\in[N], j\in\mathcal{A}_i(t): \mu_{i,j}>\mu_{i,A_i(t)}\}\right\}\mid\neg\mathcal{F}\right] \tag{8}
$$

$$
\leq\mathbb{E}\left[\sum_{t=1}^{T}\sum_{i\in[N]}\mathbb{1}\{\exists j, j'\in\mathcal{A}_i(t): \mu_{i,j}>\mu_{i,j'}, a_{j'}=A_i(t)\}\mid\neg\mathcal{F}\right]
$$

$$
\leq\mathbb{E}\left[\sum_{t=1}^{T}\sum_{i\in[N]}\sum_{j'\in\mathcal{A}_i(t)}\mathbb{1}\{A_i(t)=a_{j'}, j' \text{ is not the best arm in } \mathcal{A}_i(t)\}\mid\neg\mathcal{F}\right]
$$

$$
\leq\mathbb{E}\left[\sum_{i\in[N]}\sum_{j'\in[K]}\sum_{t=1}^{T}\mathbb{1}\{j'\in\mathcal{A}_i(t), j' \text{ is not the best arm in } \mathcal{A}_i(t), A_i(t)=a_{j'}\}\mid\neg\mathcal{F}\right] \tag{9}
$$

$$
\leq\frac{96NK\log T}{\Delta^2},
$$

where equation 8 holds since arm $a_j$ prefers $a_j$ to its matched arm $A_j^{-1}(t)$, then $a_j$ must first propose to arm $p_i$ in Algorithm 2 and thus $a_j\in\mathcal{A}_i(t)$. And equation 9 can be proved by contradiction. Suppose the matched time of $p_i$ and $a_{j'}$ is larger than $96\log T/\Delta^2$, i.e., $T_{i,j'}(t)>96\log T/\Delta^2$, then $p_i$ would not select $a_{j'}$ to match at time $t$. This is because for other better arms $a_j\in\mathcal{A}_i(t)$ with $\mu_{i,j}>\mu_{i,j'}$, if the matched time $T_{i,j}(t)$ is smaller than $96\log T/\Delta^2$, then $p_i$ would select those with fewer match times (Line 6 of Algorithm 2). And otherwise, due to Lemma B.3, $p_i$ would estimate $a_{j'}$ as sub-optimal arms and does not select it (Line 5 of Algorithm 2). □

**Lemma B.3.** *At any time slot $t$, for any player $p_i$ and arm $a_j, a_{j'}$ with $\mu_{i,j}>\mu_{i,j'}$, if $\min\{T_{i,j}(t), T_{i,j'}(t)\}>96\log T/\Delta^2$, then $\mathrm{UCB}_{i,j'}(t)<\mathrm{LCB}_{i,j}(t)$ conditional on $\neg\mathcal{F}$.*

*Proof.* By contradiction, suppose $\mathrm{UCB}_{i,j'}(t)\geq\mathrm{LCB}_{i,j}(t)$. Based on the definition of $\neg\mathcal{F}$ (equation 2) and LCB, UCB (Line 3 of Algorithm 3), it holds that

$$
\mu_{i,j}-2\sqrt{\frac{6\log T}{T_{i,j}(t)}}\leq\mathrm{LCB}_{i,j}(t)\leq\mathrm{UCB}_{i,j'}(t)\leq\mu_{i,j'}+2\sqrt{\frac{6\log T}{T_{i,j'}(t)}}. \tag{10}
$$

we can conclude

$$
\Delta_{i,j,j'}:=\mu_{i,j}-\mu_{i,j'}\leq 4\sqrt{\frac{6\log T}{\min\{T_{i,j}(t), T_{i,j'}(t)\}}}.
$$

This implies $\min\{T_{i,j}(t), T_{i,j'}(t)\}\leq 96\log T/\Delta_{i,j,j'}^2\leq 96\log T/\Delta^2$, which contradicts the fact that $\min\{T_{i,j}(t), T_{i,j'}(t)\}>96\log T/\Delta^2$. The lemma can thus be proved. □

## C  PROOF OF THEOREM 5.1

Denote $s_{\max}$ as the total number of phases of Algorithm 4 when the interaction ends. For each phase $s$, denote $t_s(\texttt{Communication})$ as the number of time slots when running the Communication

algorithm (Algorithm 5). Then when $\Delta > 0$, the regret of Algorithm 4 can be decomposed as

$$
\begin{aligned}
Reg_i(T) &= \mathbb{E}\left[\sum_{t=1}^{T}\left(\mu_{i,m_i} - X_{i,A_i(t)}(t)\right)\right] \\
&\leq \mathbb{E}\left[\sum_{t=1}^{T}\mathbb{1}\left\{\bar{A}(t) \notin M\right\}\right] \\
&\leq NK + \mathbb{E}\left[\sum_{s=1}^{s_{\max}}\left(\sum_{\tau=1}^{\ell_s}\mathbb{1}\left\{\bar{A}(\tau) \notin M\right\} + t_s(\texttt{Communication})\right)\right] \\
&\leq NK + \mathbb{E}\left[\sum_{s=1}^{s_{\max}}\sum_{\tau=1}^{\ell_s}\mathbb{1}\left\{A(\tau) \notin M\right\} \mid \neg\mathcal{F}\right] + \mathbb{E}\left[\sum_{s=1}^{s_{\max}}t_s(\texttt{Communication})\right] + \mathbb{E}\left[\sum_{t=1}^{T}\mathcal{F}\right] \\
&\leq NK + \frac{672NK\log T}{\Delta^2} + NK^2\log T + 3NK^2 + 2NK\,.
\end{aligned}
$$

where the second last inequality is due to Lemma C.1, the last inequality is due to Lemma C.2, Lemma C.3, and Lemma B.1.

If $\Delta = 0$, recall that any matching without conflicts is a stable matching as no blocking pair exists. So Algorithm 4 would only suffer regret in the index estimation phase and the Communication phase as running Subroutine-of-AE-AGS does not suffer stable regret (Lemma C.1). The regret can thus be decomposed as

$$
\begin{aligned}
Reg_i(T) &= \mathbb{E}\left[\sum_{t=1}^{T}\left(\mu_{i,m_i} - X_{i,A_i(t)}(t)\right)\right] \\
&\leq \mathbb{E}\left[\sum_{t=1}^{T}\mathbb{1}\left\{\bar{A}(t) \notin M\right\}\right] \\
&\leq NK + \mathbb{E}\left[\sum_{s=1}^{s_{\max}}\left(\sum_{\tau=1}^{\ell_s}\mathbb{1}\left\{\bar{A}(\tau) \notin M\right\} + t_s(\texttt{Communication})\right)\right] \\
&\leq NK + \mathbb{E}\left[\sum_{s=1}^{s_{\max}}t_s(\texttt{Communication})\right] \\
&\leq NK + \log T\,,
\end{aligned}
$$

where the last inequality is due to Lemma C.2.

**Lemma C.1.** *In Algorithm 4, no collision happens, i.e., $\bar{A}_i(t) = A_i(t)$ when players select arms based on the Subroutine-of-AE-AGS (Line 15).*

*Proof.* We first prove that all players maintain the same values of $\pi_{j,i}, T_{i,j}$, and $\text{Better}(i, j, j')$ for each $j, j' \in [K]$. In Algorithm 4, $\pi := (\pi_{j,i})_{j\in[K],i\in[N]}$ is determined based on which player is matched with arm $a_j$ in the corresponding time slot (Line 3-10). Since all players have the same observation, different players have the same knowledge over $\pi$. Similarly, all players have the same value of $(T_{i,j})_{i\in[N],j\in[K]}$ since they update this knowledge only when they observe that $a_j$ is matched with $p_i$ within the phase (Line 18). The comparison matrix Better is only updated during the Communication based on the selection of players in the corresponding slot (Line 14, 25 in Algorithm 5), so the value of Better among different players is also the same.

Above all, the computed matching $m$ in each time slot (Line of Algorithm 15) is the same for all players. Further based on the procedure of Subroutine-of-AE-AGS (Algorithm 2), all players are assigned with different arms. So no collision happens, i.e., $\bar{A}_i(t) = A_i(t)$ for each player $p_i$, when players select arms based on Subroutine-of-AE-AGS in Algorithm 4 (Line 15). $\square$

**Lemma C.2.** *When $\Delta > 0$,*

$$
\mathbb{E}\left[\sum_{s=1}^{s_{\max}}t_s\,(\texttt{communication})\right] \leq NK^2\log T + 3NK^2\,.
$$

*When* $\Delta = 0$,

$$\mathbb{E}\left[\sum_{s=1}^{s_{\max}} t_s\,(\texttt{communication})\right] \leq \log T\,.$$

*Proof.* We first prove the first inequality. Recall that the phase length grows exponentially until a player $p_i$ finds that an arm $a_j$ is better than $a_{j'}$ and updates its comparison flag Update_Flag as True. Based on Line 21 in Algorithm 4, the comparison information of each arm pair can only be updated once. Above all, $N$ players can update the comparison information in at most $NK^2$ phases. We can divide the total phases into several super phases where only the start phase of the super phase has length 2 and the length of all of the following phases grows. Then each super phase contains at most $\log T$ phases and there are at most $NK^2$ super phases. So the Communication procedure runs in at most $NK^2 \log T$ times.

When running Communication, one time slot would be first used for all players to transmit the Update_Flag information (Line 2-4). So the total time complexity to transmit the update flag is $NK^2 \log T$. Then players would transmit their updated pairs, with each pair costing 2 time slots and an ending slot to select nothing. Since at most $NK^2$ pairs are updated, the total time complexity to transmit the updated pairs is $3NK^2$. Thus the lemma can be proved.

When $\Delta = 0$, all players have the same preference values over all arms. Based on the definition of UCB and LCB in Line 3 of Algorithm 3, it would never happen that $\mathrm{LCB}_{i,j} < \mathrm{UCB}_{i,j'}$ for some player $p_i$ and arms $a_j$, $a_{j'}$. So the comparison information of any player would not updated in all phases. The phase length would never restart and there is only one super phase. So the total number of phases is $\log T$. And during each Communication procedure, all players only spend one time slot to transmit the Update_Flag and have no update pair to transmit. Above all, the total communication time complexity is $\log T$. $\qquad\square$

**Lemma C.3.** *In Algorithm 4,*

$$\mathbb{E}\left[\sum_{s=1}^{s_{\max}}\sum_{\tau=1}^{\ell_s} \mathbb{1}\{A(\tau) \notin M\} \mid \neg\mathcal{F}\right] \leq \frac{672NK \log T}{\Delta^2}\,.$$

*Proof.* Recall that the phase length grows exponentially if Update_Flag $=$ False and restart if Update_Flag $=$ True at the last phase. Divide the total $s_{\max}$ phases into several super-phases based on whether Update_Flag $=$ True. And denote $s_r$ as the number of phases contained in the super-phase $r$. Use $r_{\max}$ to represent the number of super-phases. For convenience, denote $\mathcal{A}_i(t)$ as the set of available arms of player $i$ at the end of Algorithm 2 when running it at round $t$. The above regret can be decomposed as

$$\mathbb{E}\left[\sum_{s=1}^{s_{\max}}\sum_{\tau=1}^{\ell_s} \mathbb{1}\{A(\tau) \notin M\} \mid \neg\mathcal{F}\right]$$

$$\leq \mathbb{E}\left[\sum_{r=1}^{r_{\max}}\sum_{s=1}^{s_r}\sum_{\tau=1}^{\ell_s} \mathbb{1}\{A(\tau) \notin M\} \mid \neg\mathcal{F}\right]$$

$$\leq \mathbb{E}\left[\sum_{r=1}^{r_{\max}}\sum_{s=1}^{s_r}\sum_{\tau=1}^{\ell_s} \mathbb{1}\left\{\exists i \in [N], j \in [K] : \mu_{i,j} > \mu_{i,A_i(\tau)} \text{ and } \pi_{j,i} \prec \pi_{j,A_j^{-1}(\tau)}\right\} \mid \neg\mathcal{F}\right]$$

$$\leq \mathbb{E}\left[\sum_{r=1}^{r_{\max}}\sum_{s=1}^{s_r}\sum_{\tau=1}^{\ell_s} \mathbb{1}\left\{\exists i \in [N], j \in \mathcal{A}_i(\tau) : \mu_{i,j} > \mu_{i,A_i(\tau)}\right\} \mid \neg\mathcal{F}\right]$$

$$\leq \mathbb{E}\left[\sum_{r=1}^{r_{\max}}\sum_{s=1}^{s_r}\sum_{\tau=1}^{\ell_s} \mathbb{1}\{\exists i \in [N], j' \in \mathcal{A}_i(\tau) : A_i(\tau) = a_{j'}, j' \text{ is not the best arm in } \mathcal{A}_i\} \mid \neg\mathcal{F}\right]$$

$$\leq \sum_{i \in [N]}\sum_{j' \in [K]} \mathbb{E}\left[\sum_{r=1}^{r_{\max}}\sum_{s=1}^{s_r}\sum_{\tau=1}^{\ell_s} \mathbb{1}\{A_i(\tau) = a_{j'}, j' \text{ is not the best arm in } \mathcal{A}_i(\tau)\} \mid \neg\mathcal{F}\right]$$

With a little abuse of notation, denote $s'$ as the first phase at the end of which $T_{i,j'} > 96 \log T/\Delta^2$. Denote $t'$ as the first round in $s'$ at the end of which $T_{i,j'} > 96 \log T/\Delta^2$. Further, denote $r'$ as the super-phase that contains phase $s'$ and $s(r')$ as the global phase index of the first phase in $r'$.

Then at any round $\tau$ that after phase $s'$, if exists better arm $a_j$ such that $T_{i,j} > 96 \log T/\Delta^2$, $p_i$ would update $\text{Better}(i, j, j') = 1$ based on $\neg\mathcal{F}$ and Lemma B.3. Subroutine-of-AE-AGS (Algorithm 2) would thus not assign $a_{j'}$ to player $p_i$. And if all of the other better arms $a_j$ have $T_{i,j} < 96 \log T/\Delta^2$, $p_i$ may still select arm $a_{j'}$ in the next phase. But recall that Subroutine-of-AE-AGS would always select the arm with the fewest selection times for $p_i$ (Line 6 of Algorithm 2), at time $t'$, the difference between $T_{i,j}$ and $T_{i,j'}$ should be no more than 1. So $p_i$ would not select arm $a_{j'}$ after the phase $s'+1$. Above all, the formula can be bounded as

$$\sum_{i\in[N]}\sum_{j'\in[K]} \mathbb{E}\left[\sum_{r=1}^{r_{\max}}\sum_{s=1}^{s_r}\sum_{\tau=1}^{\ell_s} \mathbb{1}\{A_i(\tau) = a_{j'}, j' \text{ is not the best arm in } \mathcal{A}_i(\tau)\} \,|\, \neg\mathcal{F}\right]$$

$$\leq \sum_{i\in[N]}\sum_{j'\in[K]} \left(\frac{96\log T}{\Delta^2} + \mathbb{E}\left[\sum_{\tau=t'}^{\ell_{s'}} \mathbb{1}\{A_i(\tau) = a_{j'}, j' \text{ is not the best arm in } \mathcal{A}_i(\tau)\} \,|\, \neg\mathcal{F}\right]\right.$$

$$\left. + \mathbb{E}\left[\sum_{\tau=1}^{\ell_{s'+1}} \mathbb{1}\{A_i(\tau) = a_{j'}, j' \text{ is not the best arm in } \mathcal{A}_i(\tau)\} \,|\, \neg\mathcal{F}\right]\right)$$

$$\leq \sum_{i\in[N]}\sum_{j'\in[K]} \left(\frac{96\log T}{\Delta^2} + \mathbb{E}\left[(2+4)\cdot\sum_{s=s(r')}^{s'-1}\sum_{\tau=1}^{\ell_s} \mathbb{1}\{A_i(\tau) = a_{j'}, j' \text{ is not the best arm in } \mathcal{A}_i(\tau)\} \,|\, \neg\mathcal{F}\right]\right)$$

$$\tag{11}$$

$$\leq \sum_{i\in[N]}\sum_{j'\in[K]} \left(\frac{96\log T}{\Delta^2} + 6\cdot\frac{96\log T}{\Delta^2}\right)$$

$$\leq \frac{672NK\log T}{\Delta^2},$$

where equation 11 is due to the exponentially increasing phase length. $\qquad\square$

## D   TECHNICAL LEMMAS

**Lemma D.1.** *(Corollary 5.5 in Lattimore & Szepesvári (2020)) Assume that $X_1, X_2, \ldots, X_n$ are independent, $\sigma$-subgaussian random variables centered around $\mu$. Then for any $\varepsilon > 0$,*

$$\mathbb{P}\left(\frac{1}{n}\sum_{i=1}^n X_i \geq \mu + \varepsilon\right) \leq \exp\left(-\frac{n\varepsilon^2}{2\sigma^2}\right), \quad \mathbb{P}\left(\frac{1}{n}\sum_{i=1}^n X_i \leq \mu - \varepsilon\right) \leq \exp\left(-\frac{n\varepsilon^2}{2\sigma^2}\right).$$

## E   EXPERIMENTS

In this section, we conduct a series of experiments to validate the convergence of our AE-AGS in markets with indifference and compare its performance with that of centralized ETC (abbreviated as C-ETC) (Liu et al., 2020) and phased ETC (abbreviated as P-ETC) (Basu et al., 2021), both of which can also be extended to handle indifference. In each experiment, we run all algorithms for $T = 100k$ rounds and report the averaged results over 20 independent runs. The standard errors calculated as the standard deviation divided by $\sqrt{20}$ are plotted.

To present the stable regret of each player, we first test the algorithms' performances in a small market with 5 players and 5 arms. The position of each arm in a player's preference ranking is a random number in $\{1, 2, \ldots, K\}$, similar to how the arms rank the players. Arms sharing the same position in a ranking have the same preference values, and the preference gap between two arms ranked in adjacent positions is set to $\Delta = 0.1$. The feedback $X_{i,j}(t)$ for player $p_i$ on arm $a_j$ at time $t$ is drawn independently from the Gaussian distribution with mean $\mu_{i,j}$ and variance 1. We report

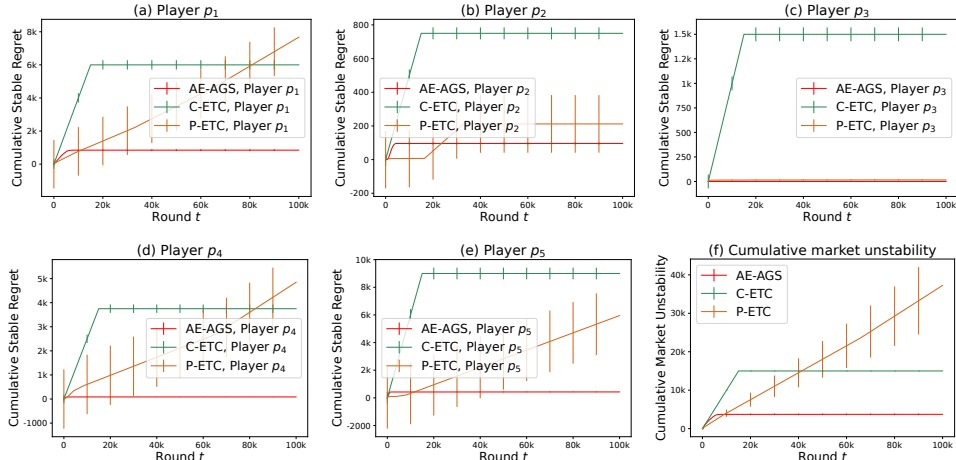

Figure 1: Experimental comparison of AE-AGS and baselines in a market with 5 players and 5 arms.

the stable regret of each player in Figure 1 (a)(b)(c)(d)(e) and the cumulative market unstability (the cumulative number of unstable matchings) in Figure 1 (f).

For generality, we also vary the value of $\Delta \in \{0.1, 0.15, 0.2, 0.25\}$ and market size $N = K \in \{3, 6, 9, 12\}$ to show the performances of algorithms. We report both the market unstability and the maximum cumulative stable regret among all players in Figure 2 (a)(c) and (b)(d), respectively.

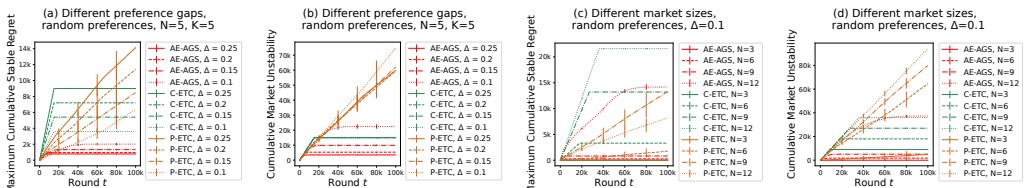

Figure 2: Experimental comparison of AE-AGS and baselines in markets with different preference gaps and market sizes.

In all tested markets, our AE-AGS consistently demonstrates good performances. This observation aligns with the theoretical results, where the performance of C-ETC is sensitive to the value of $\Delta$, performing well in markets where $\Delta$ is appropriate but worse in others. The baseline P-ETC suffers from exponential regret and has not converged within the reported horizon. The dependency of the algorithm's performance on the parameters $\Delta$ and $N$, $K$ is also consistent with the theoretical results. Specifically, as $\Delta$ decreases and $N$ or $K$ increases, the algorithm needs to pay more exploration costs, leading to higher regret.

