# OpenReview forum: "Bandit Learning in Matching Markets with Indifference"
_ICLR.cc/2025/Conference — ICLR 2025 Poster_

### Official Review · Reviewer_2KcX · 2024-10-17

**Soundness:** 2
**Presentation:** 3
**Contribution:** 2
**Rating:** 5
**Confidence:** 5

**Summary:**

The paper studies the problem of bandit learning in matching markets with ties, while previous literature usually assumed that preferences were strict. The authors propose an Adaptive Exploration Arm-guided Gale-Shapley algorithm, both in a centralized setting and a decentralized setting. The authors provide the corresponding stable regrets and conduct simulated experiments to validate the results.

**Strengths:**

The paper studies an important unexplored question of learning matching markets with indifference and indifference is common in real life applications. The authors provided both a centralized and decentralized variant of the AE-AGS algorithm, and showed the stable regrets. The authors also provide a discussion on whether other algorithms can deal with ties (Appendix A).

**Weaknesses:**

The formulation of the problem setting is not convincing enough. For matching markets with ties, there are notions of weak stability, strong stability, and super stability [1], while the authors focus on weak stability without sufficient discussions. In the example 3.1, both (weakly) stable matchings (line 221, 222) are pareto efficient matchings in the sense that no other (weakly) stable matching can pareto dominate the matching for players. Therefore, it is not accurate to state that no player-optimal stable matching exists in this sense. Finally, the definition of regret compares the difference with $m_i$, which is defined as the worst partner among all weakly stable matchings. By the definition,  {i, m_i} is not a stable matching since the worst partner might be in different matchings. I feel it might be better to define the collective stable regrets instead of individual stable regrets.

I am also concerned with the algorithmic novelty in the paper. The AE-AGS algorithm looks like a simple (not necessarily trivial) generalization of ODA algorithm [2] and the AE arm-DA algorithm [3]. These algorithms utilize arm-guided Gale-Shapley to find stable matchings and utilize UCB structure to eliminate sub-optimal arms. From my understanding, the algorithmic difference in AE-AGS is that players do not need to eliminate an arm if there are ties while still proceeding with the arm-guided Gale-Shapley. Also, the analysis and proof also look similar to [2].

[1] Robert W Irving. Stable marriage and indifference. Discrete Applied Mathematics.
[2] Fang Kong and Shuai Li. Improved bandits in many-to-one matching markets with incentive compatibility. AAAI
[3] Hadi Hosseini et. al. Putting Gale & Shapley to Work: Guaranteeing Stability Through Learning. arXiv:2410.04376

**Questions:**

I suggest the authors to formalize the problem in a more convincing way, e.g. the notion of stability, player-optimal stable matching, stable regrets (see the last part).

Questions:
(1) Can the AE-AGS algorithm be generalized to many-to-one matching markets? I'm asking this question since the ODA algorithm is designed for many-to-one matching markets.

---

> ### Author Response · Authors · 2024-11-19
> **Reply 1**
>
> We thank reviewer 2KcX for the valuable comments and suggestions. Please find our response below.
>
> -Definition of stability
>
> Thanks for pointing this out. Yes, under indifferences, [1] introduces different notions of stability including weak stability, strong stability, and super stability. Among these, strong and super stable matchings do not necessarily exist in general settings with ties. To ensure consistency with the established literature on bandit learning for matching markets, we focus on weak stability, which is the most commonly studied and applicable notion in this context. We will revise the manuscript to explicitly discuss these different notions of stability and justify our focus on weak stability more clearly.
>
> -Definition of player-optimal stable matching
>
> We would like to clarify that we follow the typical definition of player-optimal stable matching in previous bandit based works and define it as a stable matching in which every player is matched to their most preferred stable partner. In Example 3.1, such a player-optimal stable matching does not exist. We acknowledge that there are stable matchings that are Pareto-efficient, and we will add a footnote to explicitly distinguish between player-optimal stable matchings and Pareto-efficient matchings.
>
> We agree that achieving Pareto-efficient stable matchings would be a stronger and more desirable objective. However, under indifferent preferences, the exploration-exploitation trade-off becomes significantly more complex, and whether a better objective can be achieved remains an open problem. We consider this an important direction for future research.
>
> -Definition of stable regret
>
> It is correct that the worst partner for all players may not appear in a single stable matching. As analyzed in Appendices B and C, we bound the regret by bounding the number of non-stable matchings $\mathbb{E} \left[ \sum_{t=1}^T 1\left\\{\bar{A}(t) \text{ is unstable}\right\\} \right]$. Consequently, our regret guarantee for individual players is also a guarantee on cumulative market instability, which can, in a sense, be interpreted as the collective stable regret of the market. We will revise the manuscript to incorporate a discussion of this collective objective, emphasizing its role in representing overall market stability.

---

> ### Author Response · Authors · 2024-11-19
> **Reply 2**
>
> -Algorithmic novelty
>
> Although the ODA algorithm in [2] and AE arm-DA algorithm in [3] are also inspired by arm-guided DA, they differ fundamentally from our approach in exploration-exploitation design principles. Essentially, these two algorithms still adopt an "explore-then-exploit" strategy, explicitly dividing each step of the DA process. Only when the player completes an exploration step and identifies the optimal arm does the process move to the next step. Such an approach is still unable to handle indifferent preferences, which could lead to the algorithm getting stuck in one of the steps. Our key idea to address indifferences is to prevent players from facing the dilemma of determining when to stop exploration. In our approach, the available arms for players in each round are determined dynamically as the outcome of a multi-step DA process, allowing players to freely exploring. If the preferences among these arms differ, exploration will naturally conclude, and the regret can be bounded. Conversely, if preferences remain indifferent, exploration of these arms effectively becomes exploitation of the stable arm and contribute no additional regret.
>
> This key idea also highlights a fundamental difference in the motivation behind our algorithm compared to previous approaches. The motivation of [2] to adopt the arm-proposing mechanism is to prevent exploration failure caused by players directly selecting arms and being rejected under substitutable preferences. While our motivation is to eliminate the need for players to actively distinguish between exploration and exploitation phases, allowing the process to adapt dynamically.
>
> Our convergence results also set our algorithm apart from [2] and [3]. The algorithms in [2] and [3] converge to a fixed stable matching when players have strict preferences but fail to converge under indifferences. In contrast, our algorithm can guarantee stability under indifferences, with outcomes potentially switching between different stable matchings.
>
> Additionally, our proof methodology diverges significantly from that in [2]. While [2] bounds regret by analyzing the length of each discrete step in the DA process (as in Lemma 10 of [2]), our algorithm does not partition the process into distinct steps. Instead, we focus on bounding the number of non-stable matchings by constraining the occurrence of blocking pairs (as shown in our Lemma B.2).
> We will incorporate this discussion in the revised version to better clarify these distinctions.
>
> -Extension to Many-to-one setting
>
> Thank you for your question. The centralized version of our approach can naturally be extended to the many-to-one setting with substitutable preferences, as studied in [2]. However, in the decentralized setting, players would first need to estimate a unique index for communication. The current estimation process cannot be directly applied because, under substitutability, an arm can accept multiple players or reject all of them.   Moreover, players would need to learn the arms' preferences to locally execute the Subroutine-of-AE-AGS. This would require $O(K \cdot 2^N)$ time complexity as each subset of players needs to propose to each arm in the many-to-one setting. Addressing these challenges represents an interesting direction for future work.

---

> > ### Comment · Reviewer_2KcX · 2024-11-20
> > **Raising my score to 5**
> >
> > Thank you for your detailed responses. I buy your explanations for stable regret and stable matching. I also thank the authors for pointing out the algorithmic difference and the proof technique difference compared with ODA and AE arm-DA. However, I still feel that your algorithm is a simple generalization of previous algorithms and the proof technique is not much different from ODA. Therefore, I raise my score to 5.

---

> > > ### Author Response · Authors · 2024-11-21
> > >
> > > We thank reviewer 2KcX for the additional feedback and for raising the score. We would like to take this opportunity to further emphasize the contributions of our work. Our paper studies the indifference setting, which is a common scenario in real-world applications but existing methods fail to effectively handle. In this challenging context, we provide the first polynomial-time convergent algorithm, which is more robust and practical than previous approaches.
> > >
> > > From an algorithmic design perspective, while both our algorithm and the previous ODA and AE arm-DA algorithms use an arm-guided strategy, this similarity is structural rather than substantive. The key distinction lies in how we address the exploration-exploitation trade-off, which is at the heart of bandit learning algorithms. In this regard, our approach is fundamentally different and represents a significant advancement over existing methods for handling indifferent preferences.
> > >
> > > Regarding the proof technique, while there are elements that may initially appear similar, [2] places greater emphasis on the formal steps and runtime of the GS algorithm, whereas our analysis focuses more fundamentally on the blocking pairs that are the root cause of instability, thus providing a deeper understanding of the dynamic stability involved.
> > >
> > > We hope this clarification helps to resolve any remaining concerns and highlights the importance and novelty of our contributions more clearly.

---

### Official Review · Reviewer_Ragn · 2024-10-26

**Soundness:** 3
**Presentation:** 2
**Contribution:** 3
**Rating:** 6
**Confidence:** 3

**Summary:**

The authors study market stability in a two-sided market with agents whose preferences are unknown. This work allows for indifferent preferences, which have not been considered previously. They propose AE-AGS algorithm, which achieves $NK\log(T)/\Delta^2$ for each agent. Additionally, they provide numerical experiments to demonstrate their superiority over baseline methods

**Strengths:**

1. Their algorithm can achieve tight regret bound under preference indifference.
2. They also provide decentralized algorithm maintaining the tight regret bounds.

**Weaknesses:**

1. For the regret definition, they use the minimum reward value for the oracle, which may lack sufficient justification, under the absence of an optimal or pessimal stable matching.

2. In the experiments, another benchmark (C-ETC), which is much simpler, also appears to perform well in this setting.

3. The algorithm's presentation is difficult to follow. Especially, the main idea of dealing with preference indifference does not seem to be well described.

**Questions:**

1. Could you offer further justification for defining regret using the oracle's minimum value? Why minimum value is more proper than maximum reward?
2. Could you explain the key idea in your algorithm that enables it to handle preference indifference?

---

> ### Author Response · Authors · 2024-11-19
>
> We thank reviewer Ragn for the valuable comments and suggestions. Please find our response below.
>
> -Regret definition
>
> As illustrated by Example 3.1, all players cannot match with their most preferred arm in a single stable matching. So the sub-linear regret compared with the maximum reward cannot be simultaneously achieved by all players even if the algorithm converges to stable matchings. To better capture the algorithm's convergence rate toward a stable matching and to align our objective with prior works, we define regret for each player as the difference between their reward and the minimum reward across all stable matchings. As detailed in Appendices B and C, we establish an upper bound on this stable regret by bounding the cumulative number of non-stable matchings, which corresponds to cumulative market instability. Hence, our regret guarantee also provides a bound on cumulative market instability.
>
> It is worth noting that our work is the first to address indifferences and to establish a polynomial upper bound on market instability under this more general setting. We agree that a stronger objective such as the pareto-efficient matching pointed out by reviewer 2KcX would be more desirable. However, under indifferent preferences, the exploration-exploitation trade-off becomes significantly more complex, and whether a better objective can be achieved remains an open problem. We consider this an important direction for future research.
>
>
> -Performance of C-ETC
>
> Yes, C-ETC performs well in some experimental settings. However, it is crucial to highlight that this algorithm relies on the value of $\Delta$ to determine the hyper-parameter $h$ (representing the exploration budget). In our experiments, we set $h = 3000$ by testing various options (\{1000, 2000, 3000, 4000\}) across all experimental settings, selecting the smallest value that ensures convergence.  In practical applications, however, the value of $\Delta$ is unknown, and the learner cannot feasibly test multiple options to identify the best one. An inaccurate estimation of $h$ can severely impair the algorithm's performance. In contrast, our proposed algorithm does not rely on such hyper-parameters, making it more robust and practical.
>
> -The key idea to deal with indifference
>
> Existing works primarily rely on an explore-then-exploit strategy. However, under preference indifference, it becomes challenging for players to decide when to terminate exploration and begin exploitation, as they cannot discern whether two arms are truly tied or if the exploration budget is insufficient to identify their difference. The key idea of our approach is to prevent players from facing this dilemma by enabling an adaptive balance between exploration and exploitation.
>
> To achieve this, we adopt an arm-propose approach. Players continuously explore arms that propose to them while systematically eliminating suboptimal ones. For the remaining arms, if a gap exists between them, sufficient exploration will eventually distinguish the optimal choice, and the stable regret incurred by selecting suboptimal arms is sub-linear. If no gap exists, continued exploration essentially becomes equivalent to exploiting a stable arm and contributes no additional regret. This design allows our algorithm to effectively handle preference indifference while guaranteeing polynomial stable regret.

---

> > ### Comment · Reviewer_Ragn · 2024-11-20
> >
> > Thank you for your response. My concerns about the experiment and the novelty of the algorithm have been addressed. As a result, I am raising my score to 6.

---

> > > ### Author Response · Authors · 2024-11-20
> > >
> > > We thank reviewer Ragn for the response and for raising the score. We are pleased that our clarifications addressed your concerns.

---

### Official Review · Reviewer_fp6v · 2024-11-01

**Soundness:** 3
**Presentation:** 3
**Contribution:** 2
**Rating:** 6
**Confidence:** 4

**Summary:**

The authors study the problem of bandit learning in matching markets when there are ties in the users' preference over arms.  They study the stable regret, i.e. regret with respect to the least reward achieved in any stable matching. They adopt arm side proposal which leverages the fact that the arm side knows about their respective preferences. This way the user don't suffer from the dilemma of whether to declare two arms tied or continuing the exploration. Instead, the user requires only segregate non-tied arm through pairwise computations. This ensures even if ties are present a stable match is discovered with logarithmic regret.

**Strengths:**

- The authors try to study the effect of ties in the well studied field of bandit learning in matching markets.
- They design algorithms in both centralized and decentralized setting that achieves logarithmic stable regret.
- They identify that the existing algorithms with User side proposals cease to work when there is ties on the User side (the side where information is absent). The issue is identifying ties against lack of appropriate exploration.

**Weaknesses:**

- The paper lacks discussions on the user optimal regret. The arm side proposal makes it hard to obtain user optimal regret even if ties are not present.
- The motivation to move to arm side (the side that knows the preference) proposal is not clear. Is it a fundamental shift necessary for handling ties while maintaining logarithmic regret?

**Questions:**

See weaknesses.

---

> ### Author Response · Authors · 2024-11-19
>
> We thank reviewer fp6v for the valuable comments and suggestions. Please find our response below.
>
> -Player-optimal stable regret
>
> We acknowledge that, in the absence of ties, our method may not converge to the player-optimal stable matching. However, our approach is designed to be more robust to general preference scenarios compared to existing methods. The existing optimal methods [Zhang et al. (2022), Kong & Li (2023)] completely fail in the presence of indifferences (ties) and suffer $O(T)$ regret, whereas our approach is the first to address and perform well in more general indifference setting.
> We agree that a stronger objective such as the pareto-efficient matching pointed out by reviewer 2KcX would be more desirable. However, under indifferent preferences, the exploration-exploitation trade-off becomes significantly more complex, and whether a better objective can be achieved through non-arm-propose mechanisms remains an open problem. We consider this an important direction for future research.
>
> -Motivation to move the arm side proposal
>
> Recall that existing explore-then-exploit strategies fail under indifference because players cannot determine when to stop exploration, as they lack knowledge about whether indifference exists or if their exploration is insufficient. Our motivation for adopting an arm-guided GS algorithm is to address this issue by preventing players from actively managing the exploration and exploitation process. Instead, players passively select from the arms proposing to them. When preference differences exist between arms, suboptimal arms are progressively eliminated over time, allowing exploration to terminate automatically. On the other hand, when indifferences persist, alternating among these arms effectively becomes an exploitation of the stable matching, without incurring additional regret. This approach provides a robust solution to the challenges posed by indifferences, simplifying the player’s decision-making process while maintaining stability guarantees.

---

> > ### Comment · Reviewer_fp6v · 2024-11-21
> > **Response to rebuttal**
> >
> > I thank the author for clarifying my doubts in rebuttal. I am excited to see how Pareto optimal frontiers can be reached with bandit feedback with ties in future.
> >
> > It still remains unclear to me if fundamentally we need to move to arm side proposal to work with ties (Kong et al's approach of ETC is not the only possible algorithm). For example, the authors do mention Liu et al. (2020) and Basu et al. (2021) can be extended to address indifferences with user-side proposal. But agree that the last two papers are limited in their applicability.
> >
> > I will maintain my score.

---

> > > ### Author Response · Authors · 2024-11-22
> > >
> > > We thank reviewer fp6v for the further response. We agree that exploring alternative algorithms, including player-side proposals or refined arm-side proposals, to address ties and achieve stronger objectives is a valuable direction. We will consider them in the future research.

---

### Official Review · Reviewer_SXo9 · 2024-11-04

**Soundness:** 3
**Presentation:** 4
**Contribution:** 3
**Rating:** 8
**Confidence:** 2

**Summary:**

The paper is a contribution that falls under the line of works where the goal is to match participants in a market when the preferences of these participants is not known apriori. While previous works, which the paper lists, have proposed algorithms to solve this problem for the setting where each participant in the market has a strict preference order, this work proposes an approach called AE-AGS that also accounts for the case of indifferent preferences. Indifferent preference is the case when a market participant (player or bandit arm) has an equal preference among one or more options from its complementing partner.

The indifferent preference scenario is critical for real-world applications since often it is not practical or even reasonable for a market player to create a strict preference ranking order over its complementary market participant. For example, a company might be indifferent towards hiring one among a collection of equally qualified employees.

The work proposes the AE-AGS algorithm for the centralized and de-centralized with communication settings. They analyze the algorithm and provide an upper bound on stable regret for both these settings. Finally experiments are conducted to compare the approach to baselines from the literature, and establish superior performance especially in the case of preference indifference.

**Strengths:**

The work addresses an important gap in the matching markets with unknown preferences literature where previously the approaches were not able to handle the preference indifference setting but this work can. The work proposes a UCB-style algorithm (AE-AGS) for both settings when centralized decision making is feasible and provides an separate algorithm and analysis for the decentralized setting as well.

The theoretical analysis of their algorithm backed by empirical validation of their approach makes for a strong contribution overall.

**Weaknesses:**

With my level of understanding of this area, I am unable to identify any big picture weaknesses. Instead, I have presented my concerns in the form of questions.

**Questions:**

1. There appears to be some inconsistency between the message of Table 1 classifying prior work, and section 2 on related work. While the table suggests that there have been prior works (Liu et al. 2020 and Basu et al. 2021) that address the preference indifference setting however there is a line towards the end of para 2 in Section 2 that reads: "In all the above works, both players and arms are assumed to have a strict preference ranking ... ".

Please clarify whether Liu et al. 2020 and Basu et al. 2021 can indeed handle preference indifference, and if so, how your approach differs from or improves upon these prior works. This would help resolve the apparent contradiction between Table 1 and the statement in Section 2.

2. Seems to me that the definition of Stable Regret in Eqn 1 needs to be motivated better. In particular, why is stable regret bench marked against the least reward that could be obtained from a stable matching and not the maximum stable matching reward? Please provide a justification for why you chose the least reward from a stable matching as the benchmark, rather than the maximum. Additionally, please discuss the implications of this choice on your results and how the results would compare with those under alternative definitions of stable regret.

3. Please increase the font size of the legend text in Figure 1 to improve readability for printed versions.

4. Please provide a clear definition of "Cumulative Market Instability" and explain how it relates to stable regret.

5. Please clarify why enumerating all stable matchings is problematic, even for small toy problems. Also consider including results using stable regret for these smaller examples if feasible.

---

> ### Author Response · Authors · 2024-11-19
>
> We thank reviewer SXo9 for the valuable comments and suggestions. Please find our response below.
>
> -Previous approaches dealing with indifference, comparison with Liu et al. (2020) and Basu et al. (2021)
>
> We would like to clarify that all existing works assume that market participants have strict preferences. Despite these original assumptions, we carefully examined each existing approach and found that the methods proposed by Liu et al. (2020) and Basu et al. (2021) can be extended to handle indifferences with appropriate modifications to their proofs (details are provided in Appendix A). This is why we mark these two works as applicable under indifferences in Table 1. Verifying whether existing results hold under indifferences is also a contribution of our work, and we will make this clearer in the revised version.
>
> While Liu et al. (2020) and Basu et al. (2021) can be extended to address indifferences, they come with significant limitations. Liu et al. (2020) requires knowledge of the preference gap as a hyperparameter, which is a strong assumption given that players' preferences are unknown.  Basu et al. (2021), on the other hand, suffers from exponential regret growth of $O(2^{\Delta^{-2/\epsilon}})$. In contrast, our approach avoids the strong assumption of a known preference gap and achieves a polynomial regret guarantee, offering a more practical and efficient solution.
>
> -Definition of stable regret and cumulative market instability, and their connection
>
> As illustrated by Example 3.1, all players cannot match with their most preferred arm in a single stable matching. So the sub-linear regret defined compared with the maximum reward cannot be simultaneously achieved by all players even if the algorithm converges to stable matchings. To better reflect the algorithm's convergence rate toward a stable matching and align our objective with existing works, we define the regret for each player as the difference between their received reward and the least reward in a stable matching. As detailed in Appendices B and C, we upper bound this stable regret by bounding the cumulative number of non-stable matchings (i.e., cumulative market instability, $\mathbb{E}\left[\sum_{t=1}^T 1\left\\{\bar{A}(t) \text{ is unstable}\right\\}\right]$ ):
> $Reg_i(T) = \mathbb{E}\left[\sum_{t=1}^T (\mu_{i,m_i} -X_{i,A_i(t)}(t) ) \right] \le \mathbb{E}\left[ \sum_{t=1}^T 1\left\\{\bar{A}(t) \text{ is unstable}\right\\} \cdot \mu_{i,m_i} \right]  \le \mathbb{E}\left[\sum_{t=1}^T 1\left\\{\bar{A}(t) \text{ is unstable}\right\\} \right] .$
>
> Thus, our regret guarantee also serves as an upper bound on cumulative market instability which is a much stronger objective that reflects the overall market stability. Existing works in this line also adopt cumulative market instability as comparison metrics (Liu et al. (2021); Kong et al. (2022)).
>
> -Implication of the result, and comparison with previous works
>
> Though existing works does not consider indifference, we can analyze their regret under indifference and compare our result with theirs. As analyzed in Line 81-89, the state-of-the-art works (Zhang et al., 2022; Kong & Li, 2023) do not converge and suffer $O(T)$ regret under indifference. While our algorithm converges and suffer polynomial regret. As summarized in Table 1, Liu et al. (2020) requires known $\Delta$ to achieve $O(K\log T/\Delta^2)$ regret. Our algorithm removes this strong assumption with only an additional $N$ term in regret. Basu et al. (2021) suffers exponential regret $O(2^{\Delta^{-2/\epsilon}})$ which can be huge since $\Delta$ and $\epsilon$ can be small. While our result is polynomial without this exponential dependence.
>
> -Enumerating all stable matchings
>
> The previous work [1] show that enumerating all stable matchings is #P-complete and therefore cannot be solved in polynomial time if P≠NP. Suppose there are $N$ players and $N$ arms, the time complexity to enumerate all stable matchings is $O(N^N)$. Even in the small market with size $10$, this time complexity is huge. So we only report the cumulative market unstability in experiments with varing market sizes. In experiments with varying preference gaps, we additionally report the maximum cumulative stable regret among all players in Figure 2 (c) in the revised version. Our algorithm shows consistent advantage in terms of stable regret.
> [1] Robert W. Irving and Paul Leather. The complexity of counting stable marriages. SIAM Journal on Computing (1986).
>
> -Other suggestions
>
> Thanks for your other suggestions on the paper presentation. We have increased the font size of the legend text in Figure 1 in the revised version.

---

> > ### Comment · Reviewer_SXo9 · 2024-11-24
> >
> > The reviewer thanks the authors for their clarifications and updates. Will consider this in the final evaluation

---

### Meta-Review · Area_Chair_szjY · 2024-12-21

**Metareview:**

Recent studies have focused on how individuals in matching markets learn their preferences over time through repeated interactions. These markets are often modeled as scenarios where one group represents decision-makers (players) and the other as options to choose from (arms). A key challenge is minimizing stable regret to ensure equilibrium and fairness in the market. Previous research provides strong theoretical results for stable regret but assumes that every participant has clear and strict preferences. However, in real-world contexts, such as hiring or school admissions, candidates often have similar qualifications, making it difficult for decision-makers to rank them definitively. To address this issue, this work introduces a novel algorithm, adaptive exploration with arm-guided Gale-Shapley, designed to handle cases where preferences are not strictly ordered. The approach achieves robust performance with regret bounds comparable to those in strict preference settings. Experiments validate the algorithm's ability to manage these more realistic scenarios, consistently outperforming baseline methods.

The authors also adequately addressed the concerns of R-Ragn and R-2KcX re. the novelties of the proposed approach, analysis and justifications behind stable regret and stable matching.

Considering the above novelties and the additional clarification provided by the authors, I recommend accepting the paper once the authors incorporate all the reviewer's feedback in the final version.

**Additional Comments On Reviewer Discussion:**

See above

---

### Decision · Program_Chairs · 2025-01-22

Accept (Poster)